# Understanding the Spatial-Temporal Patterns and Influential Factors on Air Quality Index: The Case of North China

**DOI:** 10.3390/ijerph16162820

**Published:** 2019-08-07

**Authors:** Wenxuan Xu, Yongzhong Tian, Yongxue Liu, Bingxue Zhao, Yongchao Liu, Xueqian Zhang

**Affiliations:** 1School of Geographic and Oceanographic Sciences, Nanjing University, Nanjing 210023, China; 2Key Laboratory of Coastal Zone Exploitation and Protection, Ministry of Natural Resources, Nanjing 210023, China; 3School of Geographical Sciences, Southwest University, Chongqing 400715, China

**Keywords:** AQI, spatiotemporal analysis, influencing factors, GWR, North China

## Abstract

North China has become one of the worst air quality regions in China and the world. Based on the daily air quality index (AQI) monitoring data in 96 cities from 2014–2016, the spatiotemporal patterns of AQI in North China were investigated, then the influence of meteorological and socio-economic factors on AQI was discussed by statistical analysis and ESDA-GWR (exploratory spatial data analysis-geographically weighted regression) model. The principal results are as follows: (1) The average annual AQI from 2014–2016 exceeded or were close to the Grade II standard of Chinese Ambient Air Quality (CAAQ), although the area experiencing heavy pollution decreased. Meanwhile, the positive spatial autocorrelation of AQI was enhanced in the sample period. (2) The occurrence of a distinct seasonal cycle in air pollution which exhibit a sinusoidal pattern of fluctuations and can be described as “heavy winter and light summer.” Although the AQI generally decreased in other seasons, the air pollution intensity increased in winter with the rapid expansion of higher AQI value in the southern of Hebei and Shanxi. (3) The correlation analysis of daily meteorological factors and AQI shows that air quality can be significantly improved when daily precipitation exceeds 10 mm. In addition, except for O_3_, wind speed has a negative correlation with AQI and major pollutants, which was most significant in winter. Meanwhile, pollutants are transmitted dynamically under the influence of the prevailing wind direction, which can result in the relocation of AQI. (4) According to ESDA-GWR analysis, on an annual scale, car ownership and industrial production are positively correlated with air pollution; whereas increase of wind speed, per capita gross domestic product (GDP), and forest coverage are conducive to reducing pollution. Local coefficients show spatial differences in the effects of different factors on the AQI. Empirical results of this study are helpful for the government departments to formulate regionally differentiated governance policies regarding air pollution.

## 1. Introduction

Air pollution, which spreads rapidly and has a strong diffusive capacity, is of increasing concern worldwide because of its adverse effects on the environment, climate, and public health [1]. Because of the expansion of urban populations, the rapid development of the regional economy, and associated urbanization, North China has become one of the most polluted regions in the world [2,3]. According to statistics, the most extreme levels of PM_2.5_ in Beijing exceeded the World Health Organization (WHO) health standards 40 times in 2013 [4]. To characterize air quality more accurately and align with the WHO, the latest Chinese Ambient Air Quality Standard (CAAQ) was released in 2012, which stipulates the use of the air quality index (AQI) in a generalized way to provide timely information about air quality to the public and help citizens understand how local air quality changes over time. Haze has had a major negative impact on the physical and mental health of the public, to the point of reducing the happiness of urban residents, implying that the control of atmospheric pollution will be an ongoing major priority for North China. To strengthen the scientific management of air quality in North China, it is necessary to understand the spatial-temporal patterns and influential factors on the AQI. Clearly, it is of enormous practical value to elucidate the main influence factor of air pollution to facilitate the application of targeted control measures.

Previous studies have primarily concentrated on the variations in air quality between years, seasons, or on specific timescales in specific cities, urban agglomeration, and nationwide; thus the air pollution hotspots were identified [5,6,7]. Because the regional differences in air pollution directly reflect on the variations in the levels of socioeconomic development, air quality research in China has focused on the urbanized areas of the Pearl River Delta [8], Yangtze River Delta [9] and the Beijing-Tianjin-Hebei (BTH) Area [10]. With the increasing focus on the spatiotemporal characteristics of air pollution, the relationship between natural environmental and socioeconomic factors and air quality has attracted increasing research interest. Meteorological factors (e.g., temperature, precipitation, relative humidity, wind speed) are introduced to explain the variation of AQI and air pollutants in China [11,12]. In addition, it has been demonstrated that AQI is in response to socioeconomic factors such as urbanization, industrial structure, GDP, energy consumption, population, and technological innovations [13,14,15]. Source apportionment can also be conducted using chemical transport models (CTMs), which link the emissions of pollutants to their ambient distributions; whereas results from the models are unstable and account for the sensitive to analytical specifications, such as the number of resolved factors and pollutant input species [16,17]. Other researches has been conducted from the perspectives of the relationship between public policies and air quality improvement [18,19], air quality modelling [20], and remote sensing inversion of air pollution source intensity [21].

Although existing studies on air pollutant characteristics on different spatial and temporal scales have made substantial progress, few studies have focused on the patterns of the spatial distribution and seasonal variation of the AQI in North China, where the AQI value is extremely high. Relevant study has confirmed that air pollution presents spatial agglomeration characteristics in Chinese cities; air pollution in a city is influenced not only by itself, but also indirectly by the neighboring cities [22]. Thus multiple linear regression model ignores the spatial dependence effect and cannot accurately measure the spillover effect of the AQI. Because of the spatial dependence of urban air pollution, an increasing number of scholars have studied the spatial regression method and analyzed the main influential factors on air quality from a global perspective, which provides the basis for formulating as the joint defense for haze pollution [23,24]. However, the regression coefficients of global regression models are consistent (i.e., averaging constant) which masks the local relationships between variables. Meanwhile, there are differences in natural environment and the social and economic levels among cities in North China, and haze often presents obvious regional characteristics [25].

Therefore, there must be spatial differences in factors affecting air quality. In other words, the factors affecting haze are often heterogeneous in space [26]. The geographically weighted regression (GWR) model is a local regression model, can effectively make-up for the difficulty of quantitative interpretation of the global spatial regression model in specific spatial positions, by introducing spatial locations into the regression parameters [27]. Therefore, the GWR model can help analyze the spatial differences of the factors influencing the AQI which provides the basis for formulating relevant policies to regionally differentiate the governance of air pollution. Moreover, the temporal resolution of socioeconomic and meteorological data are often different, thus combining them in a single comprehensive research framework may provide the details of the relationship between meteorological factors and the AQI at a smaller time scale. Consequently, we take North China as a case study, collected public air monitoring data in 97 cities (71 cities are located in the study area, and 26 are located in the surrounding provinces) during January 2014–February 2017, with the aim of conducting a quantitative analysis of spatiotemporal patterns. In the influencing factor analysis, the relationship of daily meteorological factors and AQI was revealed by correlation analysis, and then both socioeconomic and meteorological factors were incorporated into the GWR framework to identify the degree of influence of each factor on AQI at the annual scale. Our paper is of important practical significance for the government to formulate regionally differentiated haze governance policies, improve the effectiveness of regional joint protection, and achieve a sustainable economic and societal development.

## 2. Materials and Methods

### 2.1. Study Area

To improve the air quality and alleviate the health burden, the Chinese government has enacted several policies, such as air pollution prevention and control (2013–2017), and comprehensive air pollution control in Beijing, Tianjin, Hebei, and surrounding areas (2017–2018) to mitigate air pollutant emissions in the North China region. Finally, in this study, 58 prefecture-level urban areas in North China which belong to Beijing Municipality, Tianjin Municipality, Hebei Province, Shanxi Province, Shandong Province, and Henan Province, are selected as the study area, taking into account the urban areas involved in above air pollution prevention plan, as well as the integrity of the administrative regions. In 2016, the total population of the region was 340 million, representing 24.8% of the population of China; in addition, the total regional GDP was 19.73 trillion yuan, accounting for 26.5% of the total for China. However, economic development and urban sprawl have generated negative environmental consequences, North China is currently considered as the most air-polluted region in China and even of the world.

### 2.2. Data Sources and Preprocessing

AQI is a dimensionless number used by government agencies to inform the public about levels of air pollution [28]. Different countries have their own air quality indices corresponding to different national air quality standards. Here, AQI was calculated based on the Chinese ambient air quality standard (CAAQS) (GB 3095-2012), released in 2012 by China’s Ministry of Environmental Protection (MEP). Calculation of the AQI involves the concentration of PM_2.5_, PM_10_, SO_2_, NO_2_, CO, O_3_ as follows:(1)$$IAQI_{p}=\frac{I_{h}-I_{l}}{B_{h}-B_{l}}\left(C_{p}-B_{l}\right)+IAQI_{l}$$

(2)$$\mathrm{AQI}=\max\left\{{\mathrm{IAQI}}_{1},{\mathrm{IAQI}}_{2},{\mathrm{IAQI}}_{3}\ldots \ldots {\mathrm{IAQI}}_{n}\right\}$$

In Equation (1), $IAQI_{p}$ is the individual AQI of pollutant item p; $C_{p}$ is the concentration of pollutant p; $B_{h}$ and $B_{l}$ respectively represent the upper and lower limits for pollutants close to $C_{p}$; $I_{h}$ represents the IAQI corresponding to $B_{h}$; and $I_{l}$ represents the IAQI corresponding to $B_{l}$. In Equation (2), *n* is the number of pollutant’s item. When the value of AQI is more than 50, the primary pollutant is air pollutant for max IAQI. AQI values vary from 0–500 and can be divided into six levels with higher levels representing greater air pollution (Table 1).

To obtain the better spatial interpolation results of AQI, three years of monitoring data were collected from 71 cities which are located in the study area, and 26 are located in the surrounding provinces (Figure 1). The data were downloaded from the website of the China Environmental Monitoring Station (http://www.cnemc.cn/). The air quality data covers the interval from 1 January 2014 to 28 February 2017, since many monitoring sites were not established before 2014, a total of 96,023 daily records were used to calculate the arithmetic mean for each city at the monthly, seasonal, and annual scales and to establish a GIS database. Herein, spring refers to March–May, summer to June–August, autumn to September–November, and winter to December–February.

On the one hand daily precipitation and wind speed data of Beijing, Tianjin, Shijiazhuang, Taiyuan, Jinan and Zhengzhou from 2014 to 2016 were collected to analyze the impact of meteorological data on AQI of typical cities (provincial capital) in North China on a daily scale. On the other hand, since the socioeconomic data obtained in this paper are all annual data, both socioeconomic and meteorological factors were incorporated into the spatial regression model framework to identify the degree of influence of each factor on the AQI at the annual scale. Combined with the relevant literature regarding factors on haze pollution and the availability of data, we selected 13 factors that have impacts on the haze pollution. These 13 factors include four meteorological factors, including temperature, precipitation, wind speed, atmospheric pressure, and nine socioeconomic factors, including annual average population, population density, gross domestic product (GDP), per capita GDP, the secondary industry as percentage to GDP, green covered area as rate of completed area, forest coverage, civilian car ownership, total gas supply. The socioeconomic data of 58 prefecture-level cities in 2015 were download from the China City Statistical Yearbook 2016. The daily meteorological data corresponding from 1 January 2015 to 31 December 2015 were sourced from the China meteorological data website which was established by the China Meteorological Information Center (http://data.cma.cn/). To keep the time dimension of the panel data involved in the modeling consistent, a spatial interpolation approach (inverse distance weighted) was used in order to generate a nationally continuous surface for these meteorological data (which included annual temperature, annual average precipitation, and annual average wind speed and annual air pressure), and zonal statistics were subsequently used in order to obtain meteorological data for each of the study cities in North China using ArcGIS software [29]. In a similar way, we employed a zonal statistical method using the administrative boundaries of the 58 prefecture-level cities in order to calculate annual average AQI for each of the study cities. We use the variance inflation factor (VIF) to determine whether there is multicollinearity between variables and Table 2 shows a brief description of the explanatory variables that were selected in this paper and results of multicollinearity test. Because the VIF of GDP are greater than 10, they discarded. Finally, the 12 × 58 variable matrix was normalized to eliminate dimension.

### 2.3. Exploratory Spatial Data Analysis

Exploratory spatial data analysis (ESDA) is based on the principle that entities with similar geographical attributes are related to each other, and that tests of the global spatial autocorrelation and local spatial autocorrelation of geographical data can be used to determine whether there is convergence or heterogeneity [30]. In spatial autocorrelation analysis, including global spatial autocorrelation and local spatial autocorrelation, the Global Moran Index (GMI) is calculated as follows:(3)$$\mathrm{GMI}=\frac{n}{\sum_{i=1}^{n}\sum_{j=1}^{n}\omega_{i,j}}\cdot \frac{\sum_{i=1}^{n}\sum_{j=1}^{n}\omega_{i,j}\left(x_{i}-\bar{x}\right)\left(x_{j}-\bar{x}\right)}{\sum_{j=1}^{n}{\left(x_{i}-\bar{x}\right)}^{2}}$$

Here, $x_{i}$ and $x_{j}$ respectively represent the AQI of city *i* and city *j*; $\bar{x}$ is the mean of x; *n* is the number of cities; and $\omega_{i,j}$ is the spatial weights matrix between city *i* and city *j*. The GMI is in the range of [−1, 1]. Values <1, =0, or >0 respectively indicate a negative correlation, no correlation, or a positive correlation between the spatial units of AQI. The significance of the global spatial autocorrelation can be determined by a statistical test which can be evaluated using the standardized statistic *Z(GMI)*, as follows:(4)$$Z\left(GMI\right)=\frac{GMI+1/\left(n-1\right)}{\sqrt{Var\left(GMI\right)}}$$
where *Var(GMI)* is the variance of *GMI*. At the 0.05 significance level, *Z(GMI)* > 1.96 indicates a positive spatial autocorrelation between the spatial units of AQI, meaning that similar high values or low values of the distribution of AQI spatial units in North China represent spatial aggregation; *Z(GMI)* < −1.96 indicates a negative correlation between AQI spatial units, and thus the units are spatially separated; and −1.96 < *Z(GMI)* < 1.96 indicates that the spatial correlation between AQI spatial units is not evident.

The Local Moran Index (LMI) can detect the correlation between an AQI space unit and its adjacent units in North China, and thus can identify spatial agglomeration or spatial heterogeneity at specific locations [31]. The LMI is defined as follows:(5)$$\mathrm{LMI}=\frac{\left(x_{i}-\bar{x}\right)\sum_{j=1}^{n}\omega_{ij}\left(x_{j}-\bar{x}\right)}{\sum_{i=1}^{n}{\left(x_{i}-\bar{x}\right)}^{2}}$$

If the LMI is positive, the map of spatial autocorrelation is distinguished: cluster of high value (HH) with high AQI and low value (LL) with low AQI. If the LMI is negative, then the spatial units with high AQI are surrounded by low AQI (HL) or that spatial units with low AQI are surrounded by high AQI.

### 2.4. Geographically Weighted Regression

Geographically weighted regression (GWR) is an improvement over the traditional regression model; it considers the autocorrelation of spatial units, embeds the geographical function of spatial data into the regression parameters, and produces the coefficient of determination (R^2^) and local regression coefficients, which change with geographical location via the observation value of adjacent spatial units. Thus, GWR is a simple but useful new technique for analyzing spatial non-stationarity which can be used to estimate a parameter at the partial scale and reveal the spatial relationship between AQI and its influencing factors. GWR is defined as follows:(6)$$y_{j}=\beta_{0}\left(u_{j},v_{j}\right)+\overset{n}{\underset{i=1}{\sum }}\beta_{i}\left(u_{j},v_{j}\right)x_{ij}+\varepsilon_{j}$$

Here, j=1,⋯,58, represents the cities in the study; $y_{j}$ represents the 58 × 1 dimensional dependent variable (yearly value of AQI); $x_{ij}$ represents the explanatory variable matrix of *n* × *k* dimensions; $\left(u_{j},v_{j}\right)$ represents the spatial coordinates of the *j*-th city; $\beta_{0}\left(u_{j},v_{j}\right)$ is the intercept for location *j*, and $\beta_{i}\left(u_{j},v_{j}\right)$ represents the *i*-th local parameter estimate for the *j*-th city; and $\varepsilon_{i}$ is an independent distribution of random errors.

GWR is calibrated by weighting all observations around a sample point using a distance decay function, on the basis of the assumption that the observations closer to the location of the sample point have a higher impact on the local parameter estimates for that location. When GWR is used, the parameters are estimated as follows:(7)$$\beta_{f}(u_{j},v_{j})=X^{T}W(u_{j},v_{j})X^{-1}X^{T}W(u_{j},v_{j})y$$
where $\beta_{f}(u_{j},v_{j})$ is the estimate of the location-specific parameter, and $W(u_{j},v_{j})$ is a diagonal *n* by *n* spatial weight matrix, the off-diagonal elements of which are equal 0 and the diagonal elements represent the geographical weight at city *j*. Here, we construct a spatial weight matrix of the geo-weighted regression model based on the Gaussian function, so that the influence of data points near *j* can be estimated with a larger weight [32]. The formula is as follows:(8)$$W_{ij}=\exp(-d_{ij}^{2}/b^{2})$$
where $W_{ij}$ is the weight of observation *j* for observation *i*, *d_ij_* is the distance between city *i* and city *j*, and *b* is the kernel bandwidth.

When the distance is greater than the kernel bandwidth, the weight rapidly approaches zero. Both fixed and adaptive kernel bandwidths can be chosen for GWR, where fixed kernel has a constant bandwidth over space, and adaptive kernel can adapt bandwidths in size to variations in data density so that bandwidths are larger in the locations where data are sparse and smaller where data are denser. We used fixed kernel bandwidth in this study, because the dispersion ranges of air pollution at different sample cities were almost identical and did not vary between cities. In addition, fixed kernel bandwidth also allows the comparison of the regression results of different samples on the same level. The optimal bandwidth was determined by minimizing the corrected Akaike information criterion (AICc) [33].

## 3. Results

### 3.1. Analysis of the Spatial-Temporal Patterns of AQI

#### 3.1.1. Spatial Variation of AQI

A total of 4384 days of “heavily polluted” weather (AQI > 200) occurred in 55 of 71 cities in the study area during the three-year study period, i.e., ~27 days per year in each city, and the annual mean AQI reached 112.6, 103.4, and 98.5, respectively, which exceeded or was close to the Grade II standard of CAAQS (Figure 2). The air quality of all provinces and cities is characterized by an improvement in the number of “excellent” and “good” ratings, with an overall decrease in the pollution rating. Spatially, the distribution of annual AQI in 2014, 2015, and 2016 in North China, divided into 10 intervals, was obtained by Kriging interpolation and the grid ratio in each interval was enumerated (Figure 3). It can be seen that the air quality in North China improved substantially from 2014–2016; the proportion of grids with annual mean AQI < 100 increased from 35.8% in 2014 to 51.9% in 2016. Specifically, annual AQI was within the range of 69–178 in 2014 and higher value (AQI > 150) were concentrated in southern Hebei; annual AQI was mainly varying from 64 to 144 in 2015, the pollution intensity of southern Heibei and Central Henan were alleviated compared to previous year; air quality further improved in 2016 with the AQI varying from 62 to 128. Although the AQI decreased year by year, the pattern of relatively high pollution still existed in the hinterland of the North China plain.

#### 3.1.2. Periodic Variations of AQI

A plot of the daily and monthly mean values of AQI of the cities in North China during 2014–2016 is illustrated in Figure 4. After reaching minimum respective yearly values in August or September, AQI begins to rise rapidly and peaks in around December. The monthly AQI is <100 during May to September, which represents the interval of better air quality within the entire year. Based on the characteristics of AQI and the air pollutant data (i.e., the levels of PM_2.5_, PM_10_, SO_2_, CO, NO_2_ and O_3_), North China can be divided into four regions by following the Ward’s method. With signal extension and application of the wavelet transform in Matlab [34], the real part of the wavelet coefficients of AQI were obtained for each region (Figure 5a). The isograms reveal the patterns of periodic oscillations and changes in the intensity and phase information of AQI at different time scales. If the real part of the wavelet is positive on a specific timescale, the change in periodicity corresponds to the position of a wave peak, indicating that the AQI was high during that period and that the air quality is poor; and when the real part of the wavelet is negative, the change in periodicity corresponds to the position of a wave trough, indicating that the AQI was low and the air quality is relatively better during that period. The greater the absolute value of the coefficient, the more substantial the periodicity. From Figure 5a it is evident that cycles of 100–400 days are the most significant in all regions and had experienced three changes in synoptic-scale oscillations that mainly occurs in the periods from late spring to early summer and from late autumn to early winter; this may be partly attributed to substantial changes in the weather during these intervals. The curve of wavelet variance was drawn by calculating the wavelet variance of the time series of AQI in each region; the period corresponding to the wave crest represents the major periodicity. The curve of wavelet variance was drawn by calculating the wavelet variance of the time series of AQI in each region; the period corresponding to the wave crest represents the major periodicity (Figure 5b). The wavelet variance curves of AQI for the four regions show similar fluctuations, indicating that their periodic characteristics are consistent. The maximum peak corresponds to a periodicity of 280 d, which is the major period of AQI oscillations in North China. In terms of the variance of the periodic oscillations, the regions are ordered as follows: Region II > Region IV > Region III >Region I, which is consistent with their relative levels of air quality and indicates that periodic variations in air quality are strongest in regions with the worst air quality. The Morlet wavelet transform was performed on the experimental data at the time scale of 300-days, and then the wavelet coefficient curve was obtained (Figure 5c). The periodic oscillation of the wavelet coefficients in the four regions is very clear at the 380-day scale, and the AQI exhibit a sinusoidal pattern of fluctuations. The peak corresponds to winter, which is generally at the end of December or the beginning of January; and the trough corresponds to summer, usually late June or early July. These results provide strong evidence for the occurrence of a distinct seasonal cycle in air pollution which can be described as “heavy winter and light summer.” It implies that government departments should focus their pollution prevention and control measures on a specific season.

The seasonal-to-interannual spatial distribution matrix of AQI from 2014–2016 is illustrated in Figure 6. Evidently, air pollution in spring, summer, and autumn decreased during the study period, and the overall size of the polluted area also decreased. However, a cause for concern is that the level of air pollution in winter is increasing, which is demonstrated by the rapid expansion of pollution in the south of Hebei and Shanxi. The seasonal mean AQI for North China is ranked as follows: winter (137.08) > spring (103.60) > autumn (95.92) > summer (80.59). Air pollution is most widespread in winter, there are large areas in southern Heibei where the quarterly average of AQI exceeded 200; in addition to northern Hebei and Shandong Peninsula, the average AQI for all areas exceeded 100. In spring, the extent of areas with AQI > 100 shrinks to the area south of Yanshan and west of Taihang Mountains, and high values of AQI appeared in central Hebei and the central Henan urban agglomeration. Evidently, on a yearly basis, the summer season has the highest air quality. Except for a few areas in southern Hebei and western Shandong, the AQI in most areas was <100 with the lowest value in Shandong Peninsula. The distribution of air pollution expands again in the autumn, with the areas of high and low values clearly separated.

#### 3.1.3. Evolution of Spatial Autocorrelation of AQI

Spatial autocorrelation analysis, including a scatter plot of Moran’s Index and an agglomeration analysis of the local indicators of spatial association (LISA), was used to analyze the temporal evolution of the spatial pattern of air pollution. Figure 7 shows GMI scatter plots for North China for 2014–2016. For 2014, a total of 47 GMI scatter values of AQI fall within the first and third quadrants (i.e., 85.5% of cities have a positive spatial autocorrelation); for 2015, a total of 64 GMI values of AQI fall within the first and third quadrants (i.e., 90.1% of cities have a positive spatial autocorrelation); and for 2016, a total of 62 GMI scatter values of AQI fall within the first and third quadrants (i.e., 87.3% of cities have a positive spatial autocorrelation). In general, the GMI increased from 0.5073 in 2014 to 0.6541 in 2016, passing the significance test at the 0.05 level, which indicates that spatial dependency of AQI in North China was enhanced, as well as the spatial spillover effects. Figure 7 also shows 71 urban space units color-coded by different types of LMI of the AQI per year. HH cities of AQI are mostly located in southern Hebei, which represents the “hot spots” of air pollution in North China. HH cities of AQI are mainly distributed in western Shanxi, northern Hebei, and Shandong Peninsula. In the past few years, the “cold spot” of air pollution in the western part of North China has gradually moved northward, indicating that air pollution in these cities has increased in recent years compared with other cities as the LL agglomeration effect weakens. It should be noted that the absence of heterogeneous units of HL and LH indicates that there is a linkage between the air pollution in the cities of North China, and that elevated AQI values in the surrounding cities will continue to enhance the local AQI value.

### 3.2. Daily Meteorological Factors on the AQI from the Statistical Analysis

#### 3.2.1. Precipitation

We screened out the rainy days of each city out and then produced a scatter plot which took the corresponding daily precipitation as the abscissa and the rate of change of AQI from previous days as the ordinate (Figure 8). The proportion of negative rates of change of AQI for the capital cities (65.25%) was greater than that for positive rates of change (32.64%) when it rained. The rate of positive and negative rates of change of AQI in each city under different rainfall conditions were also enumerated (sub-panel in Figure 8). When precipitation was <10 mm, the number of days with a negative rate of change of AQI represented 57.06%, which was slightly more than that for positive days. Thus, we infer the precipitation had some effect in flushing pollutant particles, but it is also likely to have contributed in terms of the absence of rainfall, or low rainfall, failing to dilute the pollutants, since humid air absorbs more aerosol particles. However, when precipitation exceeded 10 mm, most of the rate of change values were negative, accounting for 72.26%, indicating that the flushing action on pollutants was significant when precipitation reached a moderate level.

#### 3.2.2. Wind

The relationship between the AQI and wind speed of six capital cities in different seasons was detected by using the Pearson correlation coefficient (Figure 9). Negative correlations were the most significant in winter when the most severe pollution occurred; whereas in summer when the air quality was higher (except for the Shijiazhuang where wind speed had a significant inhibitory effect on pollutants) there was no obvious relationship between wind speed and air pollutants in the regional capitals. PM_2.5_ and NO_2_ exhibit significant negative correlations with wind speed in all four seasons; since these pollutants are mainly derived from fossil fuel combustion, increased wind speed is an important factor for accelerating the dilution and diffusion of automobile exhausts and industrial smoke. For several cities the negative correlation between wind speed and PM_10_ was not statistically significant in spring and summer; the relationship can be ascribed to warming during spring and summer which likely results in frequent stronger breezes and generate more lithogenic dust particles, which is the main source of PM_10_, leading to the suppression of the degree of negative correlation. O_3_ was always positively correlated with the wind speed (most significantly in winter), because O_3_ is the main component of smog produced by photochemical reactions of nitrous oxides and volatile organic compounds (VOC) in the atmosphere. In addition, the vertical momentum transmission will be strengthened with increased wind speed near the ground level, which is conducive to downward transport of O_3_ from high altitudes. Therefore, as the wind speed increases, the O_3_ concentration also increases. It can be seen that on inter-annual scales air pollutants are markedly negatively correlated with wind speed, indicating that increased wind speed dilutes and scatters air pollutants in North China.

Further case analysis revealed that air pollution were transported in the direction of the prevailing air flows (Figure 10). North China was affected by a westerly wind in December 2015 and by the prevailing northwesterly wind. In the process of downwind transmission, as the wind speed decreased, pollutants accumulated across the vast areas of the North China Plain located west of Taihang Mountains and south of Yanshan Mountains. The prevailing wind in January 2016 was northerly, whereas easterly winds prevailed in western Shandong Province and northern Anhui Province, causing the pollution to move southwestward; then the northern parts of Anhui and Jiangsu Provinces experienced prevailing northwesterly and southerly winds. The core area of air pollution in North China exhibits an eastward shift under the influence of the prevailing winds. In March 2016, with the onset of spring in North China, the southern monsoon began to move northward and juxtapose with cold northern air currents. Northern North China, including northern Shanxi and central and northern Heibei, experienced prevailing northwesterly winds; whereas southern North China, including Shandong, Henan and the southern part of Heibei, experienced prevailing southerly winds. Air pollution tended to move northward compared with the previous month, under the influence of the two wind directions and winds associated with local stationary air conditions. The northern part of Hebei Province was always exposed to prevailing vigorous northwesterly winds, while the Shandong Peninsula was affected by winds from the sea; consequently, both areas had better atmospheric dispersion and dilution capacity than elsewhere in North China, resulting in their consistently low AQI values. In summary, during the study period the air quality in North China was significantly affected by the monsoon; and under the influence of the prevailing wind direction, air pollutants were transported in the direction of the prevailing air flows.

#### 3.2.3. Annual Influential Factors on the AQI from the GWR Model

Before performing the GWR regression analysis, the global ordinary least square (OLS) test is performed first. Table 3 shows the basic OLS regression and we can see that the *p* value of the Joint Wald Statistic is <0.01, indicating that the model is statistically significant. Multicollinearity is not present for the independent variables when the VIF is <7.5. The standardized coefficients represents the strength and type of relationship between each explanatory variable and the dependent variable. The wind speed, secondary industry as a % of GDP, per capita GDP, forest coverage, and civilian car ownership passed the significance test; however, the remaining seven variables did not pass the significance test. At a significance level of 1%, the secondary industry as a % of GDP, civilian car ownership both have a positive impact on the AQI. However, the wind speed, per capita GDP and forest coverage have a negative impact on the AQI. The spatial autocorrelation test result of the OLS residuals shows that the Moran index of the residuals is 0.3265 and z is 4.21. This result indicates that although the R^2^ of the OLS is 0.645, the OLS results are not reliable, and there is spatial autocorrelation among the OLS residuals. Moreover, the AQIs of North China cities exhibit significant positive spatial correlations according to the above spatial autocorrelation test of the AQI. All results indicate that OLS is no longer valid because of the significant spatial spillover effects and must be introduced to build the spatial regression model.

Next, the GWR model was utilized to investigate the spatial relationship between the AQI and the variables of GWR. The optimal bandwidth was determined according to the AIC, ultimately determining the optimal bandwidth to be 319,427. The AICc is 107.6491, which is significantly smaller than that of the OLS estimation. The R^2^ is 0.75, and the adjusted R^2^ is 0.63. Both results suggest that the GWR model provides a good explanation of the spatial variation of AQI. Here, we choose representative variables which passed the significant test for analysis. The ranking of the absolute value of the average coefficient of GWR model is as follows: civilian car ownership (0.51) > wind speed (−0.45) > secondary industry as a % of GDP (0.22) > forest coverage (−0.20) > per capita GDP (−0.16). Civil vehicles and secondary industry as % of GDP had a significant positive spatial relationship with AQI, indicating that vehicle emissions and industrial structure were direct sources of air pollution. Since normalization have been taken of all the data, the coefficients represent the percentage changes in AQI due to a one percent change in the corresponding explanatory variables. Therefore, an increase of 1% in civil vehicle number and secondary industry will individually result in respective increases in AQI of 0.51% and 0.22%. Wind speed, per capita GDP, and forest coverage are significantly negatively correlated with AQI, indicating that an increase in wind speed, economic forest coverage rate are effective means of reducing air pollution on an annual scale. An increase of 1% in wind speed, per capita GDP, and forest cover will individually result in respective decreases in AQI of 0.45%, 0.22%, and 0.20%. To analyze the spatial difference of effects of different factors on the AQI more accurately and intuitively, we spatially visualize the local regression coefficients of variables (Figure 11), which pass the significance test.

Figure 11a shows the local R^2^ value varies from 0.36 to 0.88 from south to north. In Weihai of Shandong province, the local R^2^ is the highest. North Hebei, west Shanxi, and Shandong Peninsula has higher local R^2^, indicating that the spatial relationship between the AQI and influential factors in these regions is more accurate than those in the other regions. Figure 11b shows the coefficients of civil vehicle number change from south to north, with values varying from high to low. It can be seen that vehicle number in cities of Shanxi and Henan province were more positively associated with AQI than in other cities. The maximum local regression coefficient occurs in Yuncheng (0.69), followed by Linfen (0.68). Weihai held the lowest coefficient for total emissions (0.15), followed by Yantai (0.21), and Qingdao (0.27). These results indicate that the same amount of vehicle emissions can produce more pollution in inland areas than that in coastal ones, explaining the relatively dry climate is not conducive to the dilution and precipitation of pollutants from vehicle exhausts. Vehicle volume has the most significant impact on AQI, which indicates that the continuing rapid growth of car ownership in North China will lead to a deterioration in air quality. Figure 11c shows the influence of industrial structure on the air quality in different regions. The coefficients of the secondary industry proportion are positive in all cities and higher value occurs around the area Bohai, among them, Bingzhou (0.40) has the largest industrial structure local coefficient. The economies of these areas are primarily based on chemical industries and their economic growth depends largely on the contribution of secondary production, thus the proportion of secondary industry in these regions has a greater impact on the AQI compared with other regions. Figure 11d details the wind speed coefficients for the different cities in North China, demonstrating a successive decrease in these absolute value of coefficients as one moves from the northwest to the southeast. These results indicate that wind speed in the northwest, compared with southeast of study area, had a more reductive function on AQI. Specifically, the wind speed in Zhangjiakou had the greatest reduction function in relation to AQI, with a coefficient of −0.78, followed by Chengde (−0.77), and Datong (−0.76). Notably, wind speed was the only one of the four selected meteorological factors to pass the significance test, and also is the most significant factor with negative impact on AQI, which indicates that the increase of wind speed has different degrees of inhibition on AQI of cities in North China on the annual scale. Figure 11e shows green coverage has a negative impact on the AQI value in every city, indicating that increased urban landscaping can alleviate air pollution. The absolute coefficients of forest coverage change from west to east with values varying from low to high. Cities includes Weihai, Yantai, and Qingdao, in Shandong Peninsula, are the three cities with highest values; the reason is that higher levels of forest cover occur in the eastern part of North China, since the temperature and moisture conditions are more conducive to vegetation growth than in the western part of North China; thus, the degree of absorption of air pollution by the ecological system is substantially greater in the eastern part. Figure 11f shows the coefficients of per capita GDP in the 58 cities studied. The coefficients of per capita GDP were found to increase as one moves from the south to the north, only seven cities in the Henan province exerting a positive effect and remaining cities in the north a negative effect, especially in the eastern part of North China, including the coastal areas of Tianjin, Hebei, and Shandong. The maximum local regression coefficient occurs in Tianjin (−0.36). The reason for this distribution is likely that the city clusters in the coastal areas of the Bohai Sea are characterized by a high level of economic development, and that high GDP indicates that the urban residents expect a relatively high quality living environment. Notably, the Environmental Kuznets curve (EKC) assumes that there is an inverted “U-shaped” relationship between environmental pollution and per capita income. The per capita GRP of cities in North China reached $7741 in 2015, which indicated that they were entering the critical range within which there is an increased demand for improved environmental quality. It may also suggest that the government at all levels has increased its environmental protection budgets and monitoring capabilities. It is possible that in the Beijing-Tianjin-Hebei region, economic development and air pollution control measures will proceed simultaneously in the future.

## 4. Discussion

In the past few years, North China where the cities with higher AQI are concentrated in the plains, including BTH and its neighbor regions, are infamous for its serious air pollution problems [35]. Previous studies tended to focus on the spatial temporal variations of only a single pollutant, such as PM_2.5_ and PM_10_, and given the conclusion that air pollution in China is characterized by significant positive autocorrelation and higher levels in northern China and lower in southern China [36]. However, the haze pollution in North China cannot be overlooked, and haze governance has become the top priority for the North China region. Our results shows that the average annual urban AQI values during 2014–2016 reached 112.6, 103.4, and 98.5, respectively, exceeded or were close to the Grade II standard of CAAQ, the overall air quality remains poor. Spatially, given the evidence for a strengthened spatial dependence of AQI in the urban space units in this study suggested that an air pollution alleviation policies should be implemented based on both the strategies of maximizing effort and regional joint prevention and control. Such cooperation would involve joint legislation and coordinated action to monitor, assess and implement policies. Temporally, this study found that AQI in North China presents the periodical tendency and shows a sinusoidal pattern of fluctuations over the three years. What is more, although AQI in spring, summer, and autumn decreased during 2014 and 2016, the level of air pollution in winter is increasing, which is demonstrated by the rapid expansion of pollution in the southern Hebei and Shanxi. Therefore, the Chinese government should effectively control urban air pollution according to the specific location and time of the city. More efforts should be devoted to control and reduce air pollution during winter.

Many previous studies have shown that meteorological and socio-economic factors are both closely related to air quality [37,38], however, with only a few exceptions that combine them [39]. Meanwhile, since these two kinds of data have different time scales, socio-economic data are usually annual statistics and meteorological factors are recorded daily; only analyzing the influence of each factor on AQI at the annual scale will lose the details of the relationship between meteorological factors and AQI at a smaller time scale. In addition, the multiple linear regression model ignores the spatial dependence effect and cannot accurately measure the spillover effect of AQI [40]. Although the climate of the North China region is characterized by stagnant weather, which are favorable for the formation and accumulation of pollutants at the ground, we found that the air quality significantly improved when rainfall reached a moderate level (>10 mm), and the levels of air pollutants were significantly negatively correlated with wind speed at the inter-annual scale. Thus, building an efficient urban air passage structure is important in urban planning and the use of artificial precipitation can be adopted to accelerate the dilution of air pollutants during haze weather with rainfall conditions. Notably, the relevant departments also need to focus on monitoring O_3_ pollution when the wind speed increases because O_3_ was always positively correlated with wind speed.

Some studies shows that North China region has a high proportion of secondary industry, a coal-dominated energy structure, increasing traffic intensity, and central heating in winter, which aggravates local pollution [41]. However, compared with global regression models, the local regression model (GWR) can more accurately reflect the spatial differences of AQI influential factors. Result of GWR model according to our new findings shows that vehicle exhaust gases exert great influence on the AQI, especially in semi-arid region, which are inconsistent with most of the existing studies [42]. Consequently, the administration should implement stricter vehicle emissions standards and traffic restrictions on the city residents. In addition, vehicles with improved fuel efficiency, and vehicles powered by alternative energy sources (such as electric and biomass hybrid vehicles) need to be introduced. Previous researches argued that secondary industry and energy consumption can be linked to coal-burning and emission of a range of pollutants [43]. In this study, the local coefficients of GWR of industrial structure shows similar result that is positively correlated with AQI and obviously appears as regional differences. Therefore, optimizing industrial institutions, eliminating backward high-energy-consuming industries, and vigorously developing clean energy are very effective for the management of haze pollution in an industrial city. Generally, with increase in wind speed, atmospheric turbulence and convection can be expected to strengthen and provide a dynamic field for pollutant transport and spread. The local coefficients of annual wind speed shows that the inhibition effect of wind speed on AQI increased from southeast to northwest; thus, according to the spatial differences of the influences of wind speed on the AQI, relevant departments can formulate haze warning ranges, a warning time and degrees of warning more specifically. The impact of GDP on air pollution has been widely discussed and inconclusive to date. In contrast to some previous studies, which showed the positive association between the intensity of economic activities and the level of air pollution, our study showed that estimated coefficients of per capita GDP in most of the cities was negative with AQI, and especially the city clusters in the coastal areas of the Bohai Sea, with a well-developed economy. This may indicate that, to some extent, haze in North China is an inevitable consequence of rapid economic development and then turn into a declining trend as the economy continues to grow. Governments at all levels should improve their environmental protection budgets and monitoring capabilities to achieve green development. Green coverage was demonstrated to be a favorable variable for AQI level, which is in agreement with some earlier researches [44]. Our results also show that expanding forest coverage is the effective method to purify air quality, suggesting that the development ecological city or gardened city needs to be encouraged.

The potential application of this study has to be discussed. First, a single air pollutant cannot comprehensively reflect the air quality status in a place and represent the impact of air pollution on humans. Therefore, we recommend applying the comprehensive air pollutant indices to study the state of air quality. Although different countries have their own air quality indices, corresponding to different national air quality standards, public health risks increase as the AQI rises. Second, studies outside China have focused more on the relationship between AQI and human health and most analyses are conducted on the basis of traditional statistics [45,46]. Our spatial and temporal analysis pattern can be applied to study the relationship between AQI and epidemic diseases, which can help present regional differences more intuitive. Meanwhile, study of spatial and temporal variability of air pollution have relatively short period or small space units [47]. However, we believe that a more robust spatial and temporal analysis should be based on a large number of observed data. Inter-annual variation in air quality can demonstrate the effectiveness of control and mitigation strategies. Seasonal or monthly variations can result from meteorological conditions or the contribution of emission-intensive sources. In addition, the spatial heterogeneity of air quality can reflect the emission reduction efforts, urban expansion and form, land use, and other characteristics of a certain area. For these reasons, the temporal and spatial characteristics of AQI should also be analyzed in depth and thus local governments can effectively cope with air pollution according to their specific location and time. Third, the causes of haze pollution are comprehensive and complex. As far as we know, few studies have explored the driving forces of air pollution from the perspective of a combination of socio-economic and meteorological factors. Thus, the framework of estimating the contributions and spatial spillovers of different anthropogenic factors on the air quality based on GWR model can be easily applied to other regions as long as enough data is available. In this way, relevant policies can be formulated as the joint defense against haze pollution and achieve a sustainable economic and social development.

However, it is should be noted that there are several limitations in this study. There are other socioeconomic parameters, for example, employment and educational attainment, that are excluded from this paper because of the lack of data. Natural factors such as topography, temperature, and barometric pressure also require further consideration. Thus it is necessary to take more variables into account to conduct further studies. Overall, this study provides information to the general public on the spatiotemporal rules of AQI in North China and indicative function for the formulation of urban policy and improvement of air quality in North China.

## 5. Conclusions

In this paper, we employed AQI, a comprehensive index that covers and integrates six pollutants simultaneously and can reflect the comprehensive status of ambient air quality, to investigate the spatial and temporal variation of air quality in North China. Furthermore, a better understanding of nature and socioeconomic influence factors on air quality was based on the statistical analysis and ESDA-GWR model respectively, which is beneficial to policy makers in the task of formulating pollution control strategies and improving air quality. Our major conclusions are as follows.

In terms of spatial variation, the air quality in North China has improved substantially during the study interval, and the area of heavy pollution has significantly decreased. The core area of severe air pollution is located to the south of Yanshan Mountains, west of Taihang Mountains, and east of Shandong Peninsula and covers the southern parts of the municipal districts of Beijing, Tianjin, central and southern Hebei, western Shandong and northeastern Henan Province. The Global Moran Index (GMI) increased from 0.5073 to 0.6541 during the study interval, indicating that air pollution in North China became more spatially concentrated and that the contrast between the levels of air pollution between cities was intensified.

In terms of temporal characteristics, there were substantial differences between the levels of air pollution in different seasons. The application of wavelet analysis revealed that cycles with lengths of 100–400 days were the most significant in North China and in addition the amplitude of synoptic-scale oscillations varied mainly in the period from late spring to early summer and from late autumn to early winter. The maximum peak of wavelet variance of AQI corresponds to a periodicity of 280 d, represents the major period of AQI oscillations in North China and the AQI exhibits a sinusoidal pattern of fluctuations which can be described as “heavy winter and light summer.” Although air quality in spring, summer, and autumn has improved during the study period, it became worse during winter and there was a substantial expansion of the polluted area in HuangHuaiHai Plain and Fen-Wei Basin.

In terms of the principal factors influencing the air quality, the correlation analysis of daily meteorological factors and AQI are revealed as follows: The flushing effect of rainfall was significant when rainfall reached a moderate level, the relationship of AQI and air pollutants shows different degree of negative correlation at the seasonal and inter-annual scale, whereas O_3_ was always positively correlated with wind speed. Furthermore, pollutants are transmitted dynamically under the influence of the prevailing wind direction, causing shifts in their spatial location. By incorporating both socioeconomic and meteorological factors into the uniform GWR framework to identify the degree of influence of each factor on the AQI at the annual scale, showed that civil car ownership and industrial production are positively correlated with air pollution; whereas increases in wind speed, GDP, and forest cover promote improved air quality.

## Figures and Tables

**Figure 1 ijerph-16-02820-f001:**
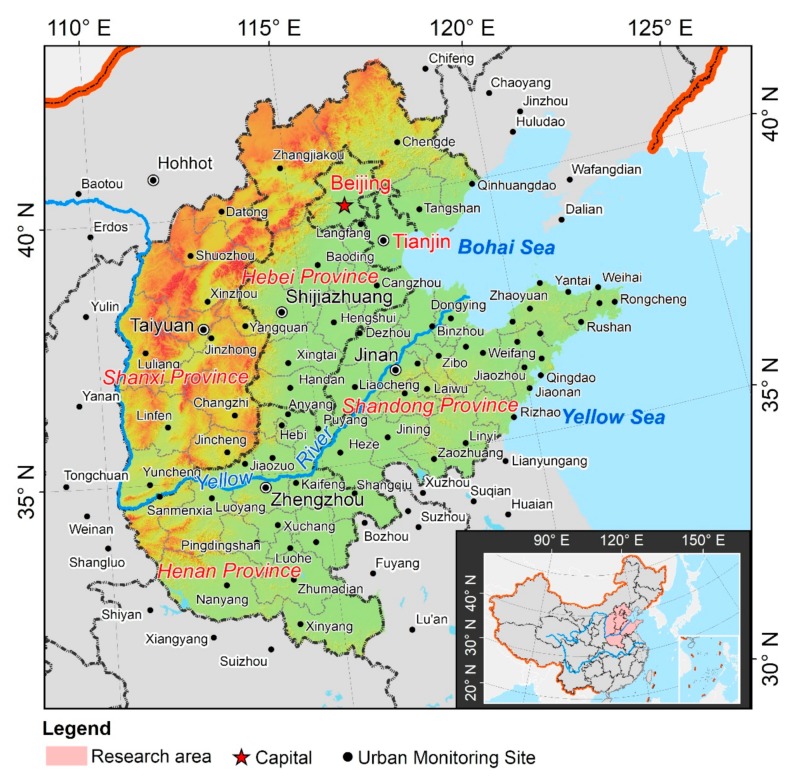
Distribution of air quality monitoring stations in the study area. Inset map shows the location of the study area in China.

**Figure 2 ijerph-16-02820-f002:**
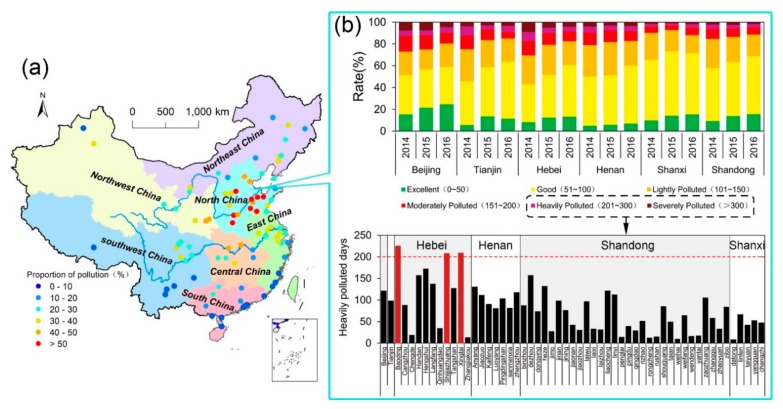
Proportions of different categories of air pollution for cities in China (2014–2016). (**a**) The AQI in North China is the highest in the country and is much higher than elsewhere in China. (**b**) The composition of AQI for each province in North China.

**Figure 3 ijerph-16-02820-f003:**
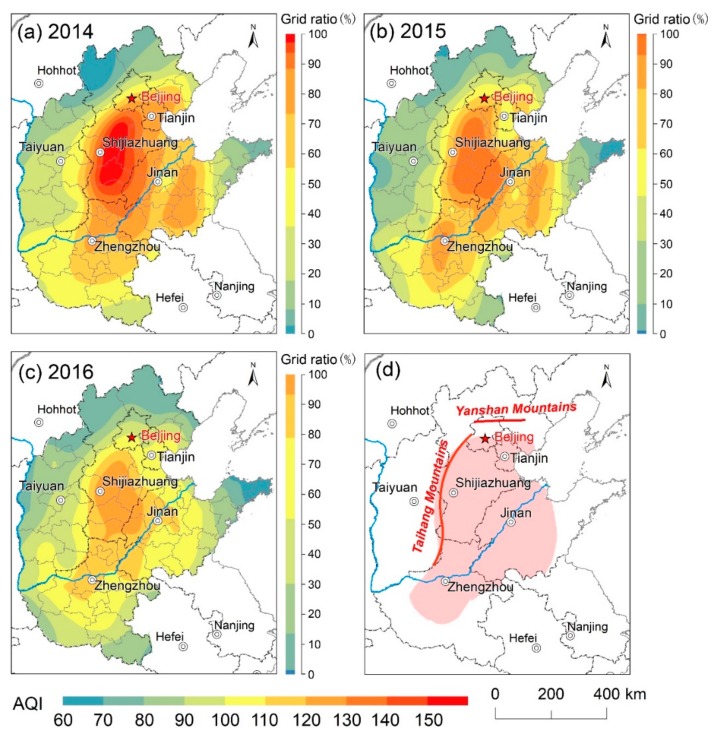
Spatial variation of AQI in North China from 2014 to 2016; (**a**–**c**) the color bars on the right of each subgraph represent the raster proportion of each AQI interval in each year; (**d**) the core area of air pollution, which was extracted using the Iterative Self-Organizing Data Analysis Technique (ISODATA), located to the south of Yanshan Mountains, west of Taihang Mountains and east of Shandong Peninsula.

**Figure 4 ijerph-16-02820-f004:**
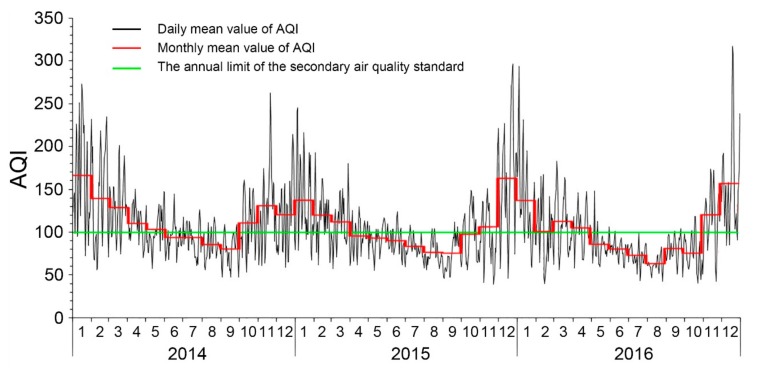
Daily and monthly average values of AQI for the cities of North China (2014–2016).

**Figure 5 ijerph-16-02820-f005:**
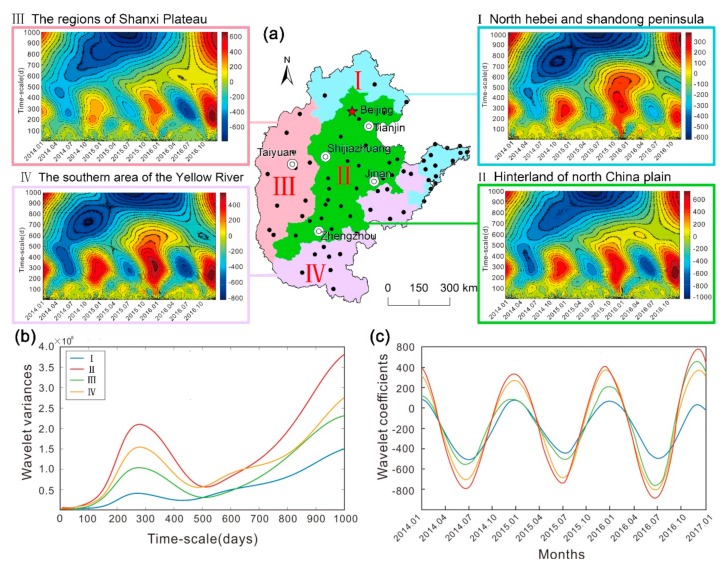
(**a**) Isograms of wavelet coefficients of AQI for four regions of North China. The abscissa is the local time (month) and the ordinate is the wavelet scale (in days). (**b**) and (**c**) are wavelet variance curves and wavelet coefficient curves of AQI of the four regions.

**Figure 6 ijerph-16-02820-f006:**
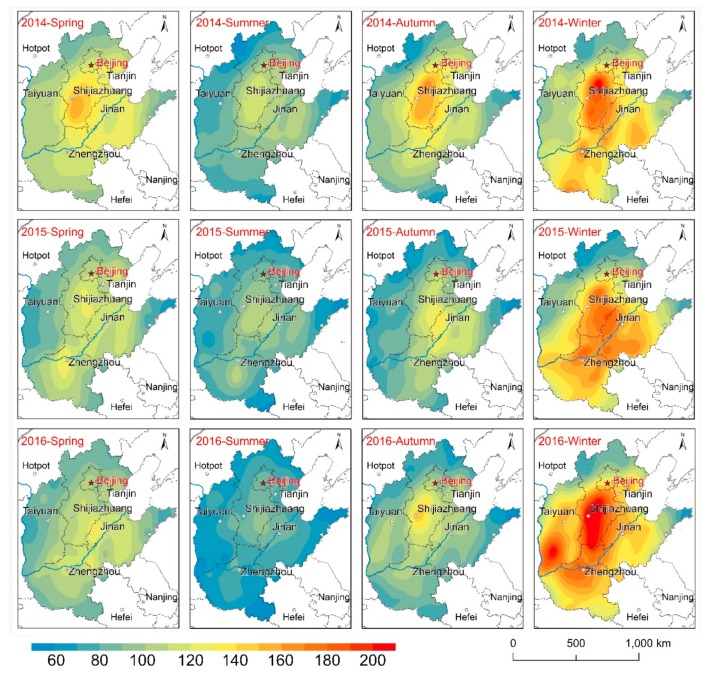
Seasonal spatial distribution matrix of AQI in North China (2014–2016).

**Figure 7 ijerph-16-02820-f007:**
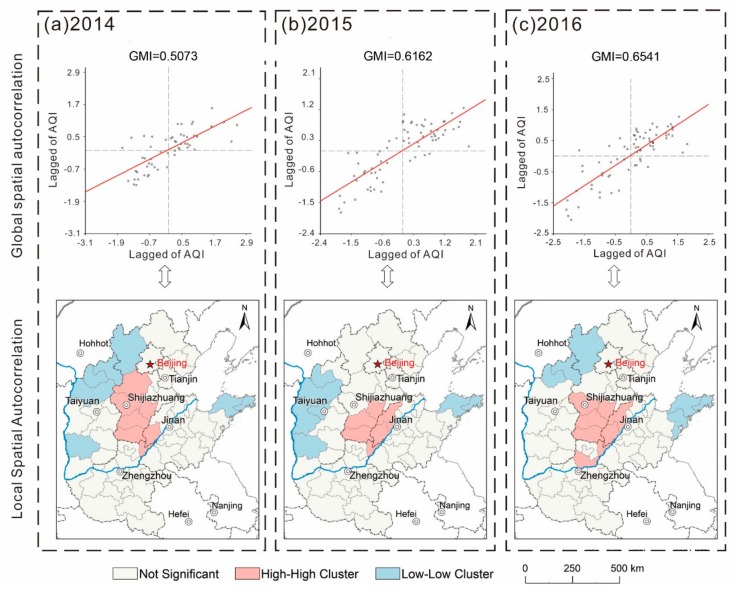
(**a**–**c**) Scatter plots of Global Moran Index (GMI) and local index of spatial association (LISA) agglomeration in North China (2014–2016). The abscissa represents the standardized AQI in cities and the ordinate represents the space lag vector which is the neighboring AQI as determined by the spatial weight weighting matrix. The GMI scatter plots are divided into four quadrants including H-H (upper right), L-L (lower left), H-L (lower right), and L-H (upper left).

**Figure 8 ijerph-16-02820-f008:**
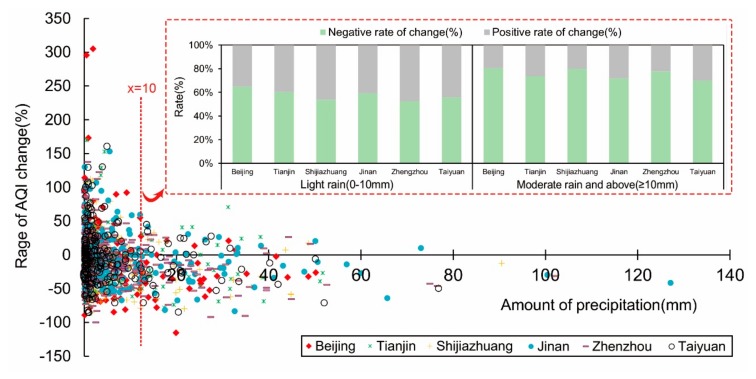
The influence of precipitation on air quality. There were 1297 rainy days in the six provincial capitals during 2014 and 2016, with light rain (0–10 mm) accounting for 78.3% of the total.

**Figure 9 ijerph-16-02820-f009:**
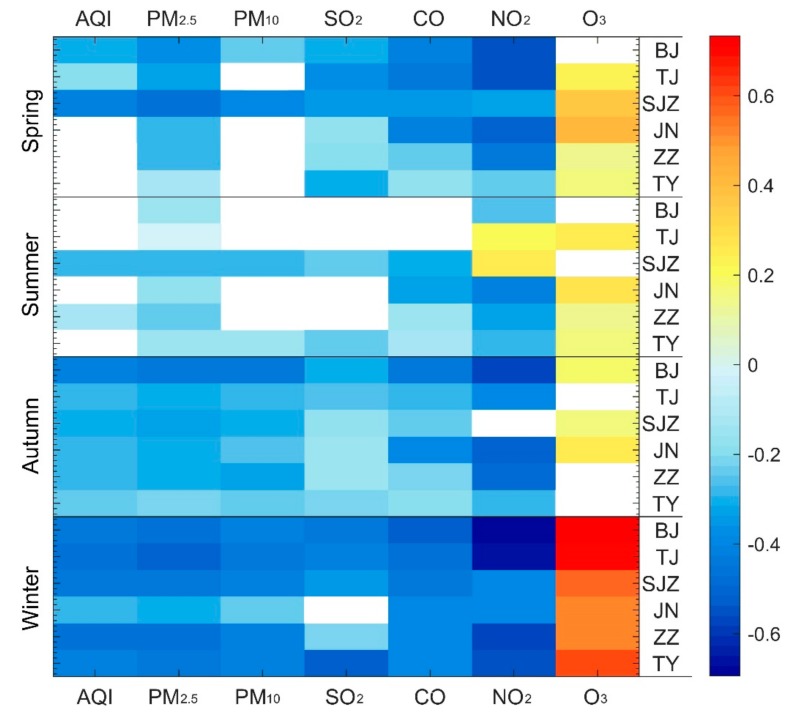
Pearson correlation coefficient of urban wind speed and AQI in different seasons for capital cities in North China. The white rectangle indicates failed passed the significance test. BJ, TJ, SJZ, JN, ZZ, TY are the abbreviation of Beijing, Tianjin, Shijiazhuang, Jinan, Zhengzhou and Taiyuan, respectively.

**Figure 10 ijerph-16-02820-f010:**
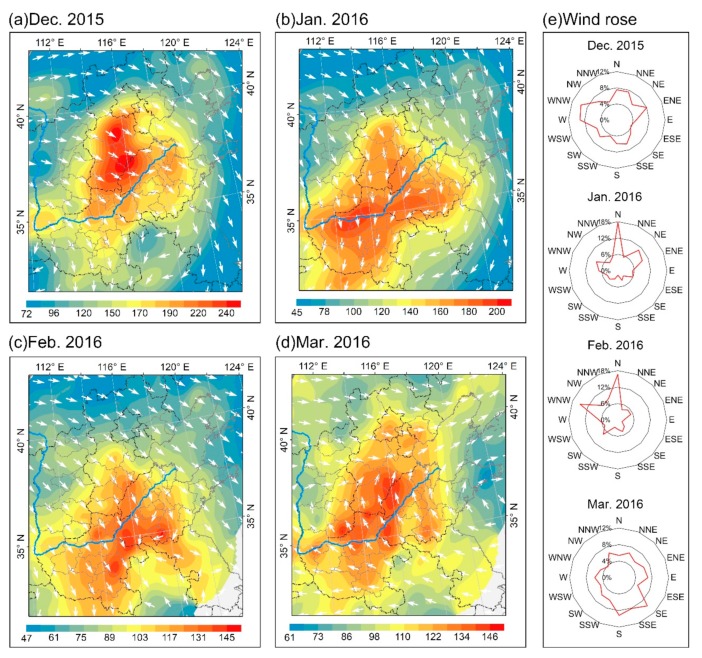
(**a**–**d**) Spatial distribution of AQI and overlay of the near ground wind field in North China from December 2015 to March 2016, and (**e**) shows wind roses showing the trajectory of maximum wind speed of major cities of North China from December 2015 to March 2016. The wind field data is derived from the National Centers for Environmental Prediction (NCEP) reanalysis data with a resolution of 2.5° × 2.5° (ftp://ftp.cdc.noaa.gov/Datasets/).

**Figure 11 ijerph-16-02820-f011:**
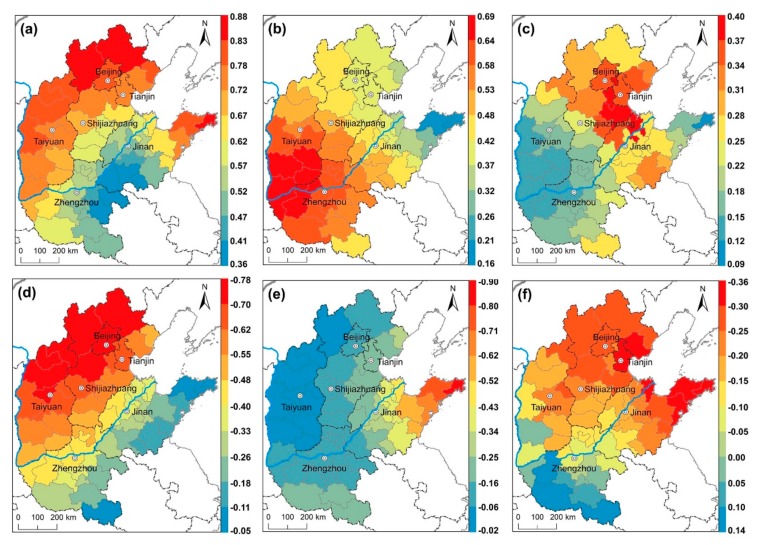
Results of the geographically weighted regression (GWR) in 58 cities. (**a**) Local R^2^ of GWR. (**b**) Local coefficients of civil vehicle number. (**c**) Local coefficients of industrial structure. (**d**) Local coefficients of wind speed. (**e**) Local coefficients of forest coverage. (**f**) Local coefficients of per capita gross domestic product (GDP).

**Table 1 ijerph-16-02820-t001:** Ranges of the air quality index (AQI) and the corresponding air pollution category.

Range	Level	Air Pollution Category	Health Implications
0–50	I	Excellent	No health implications
51–100	II	Good	Some pollutants may slightly affect very few hypersensitive individuals.
101–150	III	Lightly Polluted	Part of healthy people may experience slight irritations and sensitive individuals will be slightly affected to a larger extent.
151–200	IV	Moderately Polluted	Healthy people may manifest symptoms.
201–300	V	Heavily Polluted	Healthy people will be noticeably affected. People with breathing or heart problems will experience reduced endurance in activities.
301–500	VI	Severely Polluted	Healthy people will experience reduced endurance in activities. There may be strong irritations and symptoms and may trigger other illnesses.

**Table 2 ijerph-16-02820-t002:** Description of explanatory variables of the AQI and collinearity statistics.

	Explanatory Variables	Abbreviation	Unit	Collinearity Statistics	
				Tolerance	VIF
Meteorological factors	Temperature	Tem	°C	0.24	4.24
	Precipitation	Pre	mm	0.28	3.63
	Wind speed	WS	m/s	0.19	5.40
	Atmospheric pressure	AP	Kpa	0.30	3.37
Socioeconomic factors	Annual Average Population	AAP	10^4^ persons	0.26	3.85
	Population density	PD	person/sq·km	0.30	3.33
	GDP	GDP	10^4^ yuan	0.05	18.76
	Per capita GDP	PCGDP	yuan	0.22	4.53
	The secondary industry as percentage to GDP	SIAGDP	%	0.47	2.13
	Green covered area as rate of completed area	GCAARCA	%	0.59	1.68
	Forest coverage	FC	%	0.67	1.48
	Civilian car ownership	CCO	One car	0.16	6.01
	Total Gas Supply	TGS	10^4^ m^3^	0.15	6.64

**Table 3 ijerph-16-02820-t003:** Overall results of OLS test.

Variable	Standardized Coefficients	*p*	VIF	Std. Error	t-Statistic
Air pressure	0.1896	0.1363	3.3489	0.1444	1.313405
Temperature	−0.1844	0.1808	4.1055	0.1598	−1.153739
Wind speed	−0.5833	0.0013 *	5.0591	0.1774	−3.286439
Precipitation	−0.1862	0.2021	3.5785	0.1492	−1.247577
Population	−0.0820	0.5102	2.7766	0.1314	−0.623908
Population density	0.0866	0.5464	3.0296	0.1373	0.631044
Secondary industry as a % of GDP	0.1772	0.0944 *	2.1108	0.1146	1.545896
Per capita GDP	−0.1855	0.0044 *	2.2576	0.1185	−1.5649
Green covered area as rate of completed area	0.0714	0.3241	1.3345	0.0911	0.783747
Forest coverage	−0.2272	0.0085 *	1.3858	0.0928	−2.446134
Total gas supply	−0.3045	0.2167	3.5997	0.1497	−2.034404
Civilian car ownership	0.6319	0.0005 *	5.1361	0.1788	3.533406
Joint Wald Statistic	248.7073	0.0000 *			
AICc	127.5710				
Adjusted R2	0.6450				

* The values with gray shading passed the significance test.

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
