# Peer review of "Understanding the Spatial-Temporal Patterns and Influential Factors on Air Quality Index: The Case of North China"

_ijerph, 2019, doi:10.3390/ijerph16162820_

Round 1
Reviewer 1 Report
Comments on ijerph-539215 This paper looks at the spatiotemporal patterns of air quality and its factors in North China with local spatial statistical models. I think the first half of the paper is good, describing the spatial and seasonal characteristics of air quality and highlighting the potential correlations with meteorological factors. However, the part of influencing factor analyses of air quality is not convincing. 1. The primary research design flaw is the missing of meteorological factors in the OLS and GWR modelling analyses. Given precipitation and wind (especially the prevailing wind direction and strength) are demonstrated to be potential factors of air quality, they should be included in the local spatial statistical models. 2. I am not sure the part of global and local spatial autocorrelation analyses of air quality is useful. The motivation is not clearly stated. 3. In terms of GWR modelling, how the temporal dimension is captured in the analysis. This needs to be clarified. 4. Some robustness checks on the GWR modelling results are expected, for instance the use of different weighting schemes and bandwidth parameter selection methods. This matters if you want to draw a valid and convincing conclusion on the potential factors of air quality. 5. Given that the wind could affect air quality, it would be useful to consider the air pollution levels out of the study areas. That is, how air quality (and meteorological factors) outside of the study area could affect the air quality of cities under study. Line 72. It is not fair to say that traditional regression analysis is based on geospatial homogeneity hypothesis and ignoring the spillover effects of air pollution. This could be accounted for even in a tradition non-spatial regression framework. Not sure about the spatial autocorrelation analysis of AQI. Air quality out of the air; wind Rain seems to have non-linear effects on AQI,
Reviewer 2 Report
Regarding the article entitled " Understanding the Spatial-temporal Patterns and influential factors on Air Quality Index: The Case of North China" my judgment is positive: Accept for publication with minor revisions
This paper approaches the issue of inter-region transport of air pollutants and their spatial and temporal variability in a comprehensive, alternative way. It helps to integrate our understanding of the influential factors of atmospheric chemistry and transport. Reported are very interesting results, and altogether, I have enjoyed reading this manuscript.
The research question was investigated using various data analytics techniques (namely exploratory spatial data analysis, geographically weighted regression, correlation analysis, wavelet transform, multiple linear regression) and all the assumptions were empirically investigated. All results and conclusions were substantiated with appropriate metrics and visualizations.
Some minor comments are:
- The paper lacks a comment on Chemical transport models (CTMs). CTMs are computer models that link the emissions of pollutants to their ambient distributions. They integrate the meteorological, chemical, and physical processes that control the fate and transport of pollutants in the atmosphere. I think that this paper succeeds to isolate the impact of certain meteorological factors on air pollution, differentiating thus its contribution from the established CTMs.
- Indices and quantities within formulas (3)-(6) should be given and explained in detail. What I in formula (4) stands for? What m,n in formula (5) stand for?
- The results of stepwise multiple linear regression mentioned in line 144 should be presented in the text or in an appendix.
- In figure 3 the color legend is not the same in the three geographical heat maps. If the visualizations aim at visual comparison, the same color scale should have been used.
- In line 276 I think that GML should be corrected to GMI
- In line 372 the phrase “Factors have no multiple linearity” should be corrected to “Independent variables are not highly correlated” or “multicollinearity is not present for the independent variables”. The term “factors” should not be used, as is related and refers to factor analysis.
- Sotware used for the analysis should be cited and listed in the References section.
Reviewer 3 Report
Topic addressed by the paper is interesting. A few suggestions: 1. Explain why the study focused on selected urban areas 2. In terms of meteorological parameters, the analysis has to look more carefully into wind direction, and surrounding land use in determining the sources responsible for the AQI. Regional transport is another factor to be evaluated. 3. It might be useful to evaluate the trends with the traffic activity data to see if there is any correlation 4. The potential application of this study has to be discussed more in detail
Round 2
Reviewer 1 Report
All the comments have been carefully addressed in the revised manuscript and I have no further issues